# GLIM Criteria Using Hand Grip Strength Adequately Predict Six-Month Mortality in Cancer Inpatients

**DOI:** 10.3390/nu11092043

**Published:** 2019-09-01

**Authors:** Victoria Contreras-Bolívar, Francisco José Sánchez-Torralvo, María Ruiz-Vico, Inmaculada González-Almendros, Manuel Barrios, Susana Padín, Emilio Alba, Gabriel Olveira

**Affiliations:** 1Unidad de Gestión Clínica de Endocrinología y Nutrición, Hospital Regional Universitario de Málaga/Universidad de Málaga, Instituto de investigación Biomédica de Málaga (IBIMA), 29007 Málaga, Spain; 2Unidad de Gestión Clínica de Oncología Médica, Hospital Regional Universitario de Málaga/Universidad de Málaga, 29010 Málaga, Spain; 3Unidad de Gestión Clínica de Radiodiagnóstico, Hospital Regional Universitario de Málaga, 29010 Málaga, Spain; 4Unidad de Gestión Clínica de Hematología y Hemoterapia, Hospital Regional Universitario de Málaga, 29010 Málaga, Spain; 5Instituto de investigación Biomédica de Málaga (IBIMA), 29010 Málaga, Spain

**Keywords:** oncology, subjective global assessment, GLIM criteria, malnutrition, hand grip strength

## Abstract

Protein-calorie malnutrition is very frequent in cancer patients and is associated with an increase in morbidity and mortality. Recently, the Global Leadership Initiative on Malnutrition (GLIM) criteria were proposed to standardize the diagnosis of malnutrition. Nevertheless, these criteria were not validated in prospective studies. Our objective is to determine the prevalence of malnutrition in cancer inpatients using different diagnostic classifications, including GLIM criteria, and to establish their association with length of stay and mortality. Hence, we designed a prospective study. Within the first 24 hours of admission to the Inpatient Oncology Unit, subjective global assessment (SGA) was carried out, and anthropometric data (body mass index (BMI), mid-arm circumference (MAC), arm muscle circumference (AMC), fat-free mass index (FFMI)) and hand grip strength (HGS) were obtained to assess the reduction of muscle mass according to GLIM criteria. Length of stay, biomarkers (albumin, prealbumin, C-reactive protein (CRP)), and in-hospital and six-month mortality were evaluated. Regarding the 282 patients evaluated, their mean age was 60.4 ± 12.6 years, 55.7% of them were male, and 92.9% had an advanced-stage tumor (17.7% stage III, 75.2% stage IV). According to SGA, 81.6% of the patients suffered from malnutrition (25.5% moderate malnutrition, and 56.1% severe malnutrition), and, based on GLIM criteria, malnutrition rate was between 72.2 and 80.0% depending on the used tool. Malnourished patients (regardless of the tool used) showed significantly worse values concerning BMI, length of stay, and levels of CRP/albumin, albumin, and prealbumin than normally nourished patients. In logistic regression, adjusted for confounding variables, the odds ratio of death at six months was significantly associated with malnutrition by SGA (odds ratio 2.73, confidence interval (CI) 1.35–5.52, *p* = 0.002), and by GLIM criteria calculating muscle mass with HGS (odds ratio 2.72, CI 1.37–5.40, *p* = 0.004) and FFMI (odds ratio 1.87, CI 1.01–3.48, *p* = 0.047), but not by MAC or AMC. The prevalence of malnutrition in advanced-stage cancer inpatients is very high. SGA and GLIM criteria, especially with HGS, are useful tools to diagnose malnutrition and have a similar predictive value regarding six-month mortality in cancer inpatients.

## 1. Introduction

Disease-related malnutrition is an alteration of intake and/or assimilation of nutrients, which leads to changes in the body composition and reduced functional capacity [1]. Inflammation promotes malnutrition, being associated with the presence of anorexia, reduced intake, altered metabolism, increased energy expenditure, and increased muscular catabolism and sarcopenia, which lead to low fat-free mass and reduced functional capacity [1,2,3,4]. 

Clinical diagnosis provides a simple approach to the recognition of chronic or mild recurrent inflammation, which is likely to be associated with malignant disease. Cancer disease and nutritional status are closely linked as the symptoms caused by the disease, the associated secondary complications, and the antineoplastic therapies increase the risk of malnutrition [5,6]. 

Malnutrition and cancer cachexia are common among these patients, being present in up to 80% of them and increasing morbidity and mortality [5]. Notwithstanding, this process can be accompanied by a normal or high body mass index (BMI) [7], which supports the importance of performing a proper nutritional assessment. Nevertheless, there is no “gold standard” for determining nutritional status and most current nutritional assessment techniques are based on their ability to predict clinical outcomes [1,6,8]. Recently, the Global Leadership Initiative on Malnutrition (GLIM) criteria for the diagnosis of malnutrition were published with the aim to build a global consensus around core diagnostic criteria for malnutrition in adults in clinical settings [4]. In case of nutritional risk, these criteria recommend performing a nutritional assessment evaluating phenotypic (unintentional weight loss, low BMI, and/or reduced muscle mass) and etiologic criteria (reduced intake or assimilation, and/or inflammatory response). To diagnose malnutrition, at least one phenotypic criterion and one etiologic criterion should be present. To estimate fat-free mass, the consensus proposes several techniques that were validated, such as dual-energy X-ray absorptiometry, bioelectrical impedance analysis, computed tomography, or magnetic resonance imaging. Nonetheless, these techniques may not be available at the bedside. As a result, the consensus recommends replacing them by physical examination or standard anthropometric measures like mid-arm muscle or calf circumferences. Furthermore, other techniques for functional assessment like hand grip strength (HGS) could be considered as a supportive measure. In cancer patients, muscle weakness and fatigue are particularly frequent and are associated with increased morbidity and mortality [9]. 

After the launch of the GLIM consensus, it is now important to use the criteria in prospective cohort studies in order to validate their relevance for clinical practice and to determine their capacity to predict adverse clinical outcomes. In this sense, it is interesting to evaluate their association with mortality in cancer inpatients, as well as to compare these criteria with other validated tools like subjective global assessment (SGA) [8,10], and to assess the use of different techniques for measuring body composition, including skinfold thickness measurement or HGS. 

Our hypothesis is that GLIM criteria, using simple bedside-available tools for measuring muscle mass, can adequately predict six-month mortality in cancer inpatients in a similar manner to other widely used tools, like SGA, that are gold standards for other authors [8,11].

On the basis of this, the objective of our study was to determine the prevalence of malnutrition according to SGA and GLIM criteria and to determine which nutrition-related classification better predicts six-month mortality in cancer inpatients.

## 2. Materials and Methods

We performed an observational, prospective study of clinical practice between October 2017 and April 2018. In total, 351 patients were admitted to the Inpatient Oncology Unit at the Hospital Regional Universitario de Málaga. We considered the following inclusion criteria: diagnosis of neoplasm (without distinguishing cause of admission, pathology, or age), estimated length of stay ≥48 hours, and signing the informed consent. The exclusion criteria were as follows: estimated length of stay <48 hours, readmission before 30 days, actively dying, and lack of informed consent. Finally, 282 patients were evaluated, and 69 were excluded (Figure 1).

### 2.1. Assessment of the Cancer Status

The following variables were considered: type of neoplasm, tumor stage, Charlson comorbidity index, antineoplastic treatment, and cause of admission.

### 2.2. Assessment of the Nutritional Status

Within the first 24 hours after admission, the following tests were performed: nutritional screening according to the malnutrition universal screening tool (MUST) [12], subjective global assessment [13], and fasting blood collection (blood count, albumin, prealbumin, and C-reactive protein (CRP)); these data were used to calculate CRP/albumin ratio and the Glasgow prognostic score, a systemic inflammation-based scoring system [14]. 

### 2.3. Malnutrition According to GLIM Criteria

#### 2.3.1. Phenotypic Criteria

We assessed unintentional weight loss (>5% in 6 months), low BMI (for age <70 years, normal values were considered as BMI ≥20 kg/m^2^; for age ≥70, normal values were established as BMI ≥22 kg/m^2^), and/or reduction of muscle mass based on four possible criteria: fat-free mass index (FFMI), hand grip strength, mid-arm circumference (MAC), and arm muscular circumference (AMC). To this effect, the following anthropometric measures were obtained: weight, height, and BMI. Height was calculated at baseline with a stadiometer (Holtain Limited, Crymych, U.K.), and weight was calculated with a weighing scale adjusted to 0.1 kg (SECA 665, Hamburg, Germany). 

MAC was measured using a flexible and non-elastic tape. This value and triceps skinfold were used to estimate AMC. A lower value than the fifth percentile (p5) was considered low muscle mass [15]. 

Measurement of triceps skinfold was performed in triplicate by the same investigator using a Holtain constant pressure caliper (Holtain Limited, Crymych, U.K.) in the dominant limb, and the mean was calculated according to the recommendations of SEEN (Sociedad Española de Endocrinología y Nutrición) [16,17]. Percentages and kilograms of fat mass and FFM were estimated according to the formulas of Siri and Durnin, and Womersley [18,19]. For FFMI, the cut-off points established by the European Society for Clinical Nutrition and Metabolism (ESPEN) were applied, considering low muscle mass for values <15 kg/m^2^ in women and <17 kg/m^2^ in men [1]. 

Hand grip strength was measured in the dominant hand with a Jamar dynamometer (Asimow Engineering Co., Los Angeles, CA, USA). For this test, the patients were sitting comfortably with shoulder adducted and forearm neutrally rotated, elbow flexed to 90°, and forearm and wrist in a neutral position. They were told to perform three consecutive contractions one minute apart from each other, and the mean value was calculated. Results were expressed in absolute figures, and values under the fifth percentile of the Spanish normative reference data [20] were considered as low strength. 

#### 2.3.2. Etiologic Criteria

We assessed the following etiologic criteria: reduced intake (estimated as per quartiles) or assimilation (as per clinical record), and/or inflammatory response of the disease (chronic disease-related inflammation was evaluated using Glasgow prognostic score) [4,14]. To diagnose malnutrition, at least one phenotypic criterion and one etiologic criterion should be present [4]. 

### 2.4. Data Analysis

Quantitative variables were expressed as means ± standard deviation. Comparison between qualitative variables was performed using a chi-square test, with Fisher correction when necessary. Quantitative variable distribution was assessed using a Kolmogorov–Smirnof test. Differences between quantitative variables were analyzed using Student’s *t*-test and, for variables not following a normal distribution, using non-parametric tests (Mann–Whitney or Kruskall–Wallis). We designed multivariate logistic regression models in which the dependent variable was six-month mortality according to the various classifications of nutritional status studies (SGA, GLIM criteria using hand grip strength, FFMI, MAC, or AMC) controlling also for sex, age, and tumor stage. For calculations, significance was set at *p* < 0.05 for two tails. The data analysis was performed with the SPSS 22.0 program (SPSS Inc., Chicago, IL, USA, 2013).

### 2.5. Ethics

The study was approved by the Provincial Research Ethics Committee of Málaga, and all participants signed the informed consent. The ethical principles stated in the latest revision of the Declaration of Helsinki and good clinical practice standards were applied. 

## 3. Results

In total, 282 patients admitted to Inpatient Oncology Unit were evaluated. Their mean age was of 60.4 ± 12.6 years, and 55.7% of them were male. Their general features are displayed in Table 1. Most patients (92.9%) had an advanced-stage tumor (17.7% stage III, 75.2% stage IV), and the most frequent types of neoplasm were lung (25.2%), colon (13.0%), breast (13%), and esophagogastric (11.8%). At the moment of their admission, a nutritional status screening according to MUST was performed, detecting malnutrition risk in 82.9% (234) of patients: 14.9% moderate risk, and 68.1% high risk (Table 2). According to SGA, 81.6% (230) of patients presented malnutrition, and, according to GLIM criteria, malnutrition was detected in 72.2–80% depending on the tool used (Figure 2). We further detected low BMI in 20.6% (58) of the patients and low fat-free mass in 42.2% (119) of them, according to ESPEN criteria. Moreover, 95.4% of patients presented disease-related inflammation (Glasgow prognostic score >0). After six months, 47.9% (135) of the patients were deceased.

Charlson comorbidity index, CRP/albumin ratio, and length of stay values were significantly higher among malnourished patients, and their values of BMI, albumin, and prealbumin were significantly lower compared with normally nourished ones, with all the diagnostic tools used. In addition, in-hospital mortality and six-month mortality were significantly higher among malnourished patients in assessments according to SGA and GLIM criteria using HGS and fat-free mass index, but not using MAC or AMC. Readmissions also tended to be higher among malnourished patients, although with no statistical significance (Table 3). 

Table 4 shows the logistic regression data (crude and adjusted) for the risk of death at six months for different nutritional-assessment methods. After adjusting for confounding variables like tumor stage and age, an increased risk of mortality was significantly associated with severe malnutrition by SGA (odds ratio (OR) 2.73; 95% confidence interval (CI) 1.35–5.52) and malnutrition by SGA including both moderate and severe malnutrition (OR 2.23; 95% CI 1.14–4.38). There was also a significant increase in six-month risk of mortality in patients with malnutrition according to GLIM criteria using and hand grip strength (OR 2.72; 95% CI 1.37–5.4) and using FFMI (OR 1.87; 95% CI 1.01–3.48). A non-significant increase trend in mortality was found in malnourished patients according to GLIM criteria using MAC and AMC.

## 4. Discussion

To the best of our knowledge, no prospective studies in cancer inpatients validated GLIM criteria using simple and bedside-available methods like anthropometry or dynamometry. Based on our results, both SGA and GLIM criteria, especially with the use of HGS, are useful, simple, and available tools for the diagnosis of malnutrition. Furthermore, they have a similar predictive value for estimating six-month mortality in cancer inpatients.

In our study, the prevalence of malnutrition was very high according to SGA, a simple, safe, and inexpensive tool validated for diagnosing malnutrition in diverse patient populations, including cancer [21,22]. It has an adequate inter-observer concordance (if it is performed with the proper training) and enables clinical decisions to be made at the bedside with no need for laboratory variables or complex body composition assessment techniques [4,8]. Malnutrition according to SGA is associated with worse quality of life, higher length of stay, more complications, and is an independent predictor of overall survival in cancer patients [23,24]. With this tool, malnutrition was diagnosed in 81.6% of patients, being quite similar to the other used tools and the reported bibliography.

Using GLIM criteria, malnutrition was detected in 72.2–80.0% of patients, depending on the tool used for the assessment of muscle mass. These malnutrition prevalence data are consistent with previous publications.

GLIM criteria for the diagnosis of malnutrition include the assessment of phenotypic (unintentional weight loss, low BMI, and/or reduced muscle mass) and etiologic criteria (reduced intake or assimilation, and/or inflammatory response) [4]. Validity is well established for unintentional weight loss [11]. Therefore, it must be a priority to obtain repeated weight measures over time to identify trajectories of decline, maintenance, and improvement. BMI was proven to be an independent predictor of survival in patients with cancer [5]. Nevertheless, low BMI cannot be used as a sensible marker of nutrition status. Currently, people are often overweight or obese and would need to lose a substantial amount of weight before presenting low BMI. In our sample, even if BMI was lower in patients with malnutrition using different assessment methods, only 20.6% of subjects were below the established cutoff point [4,5], indicating the possible high prevalence of sarcopenic obesity in our series. 

Patients with cancer may have lower FFMI and less strength than healthy controls. There is no consensus on the best way to measure and define reduced muscle mass. As an alternative when the recommended methods (dual-energy X-ray absorptiometry, bioelectrical impedance analysis, computed tomography, etc.) are not available, physical examination or anthropometric methods can be used and are low-cost and user-friendly. In our study, applying GLIM criteria, we found a prevalence of malnutrition of 72.2% assessing MAC, 71.8% assessing AMC, and 77.6% with FFMI, which makes them consistent alternative measure to assess muscle mass. 

In situations in which muscle mass cannot be assessed, muscle strength is an appropriate supporting proxy. Reduced strength assessed by dynamometry (hand grip) was strongly correlated with the presence of post-surgery complications [25,26], longer hospital stays [27], reduced functional capacity, and decreased survival rate in other studies [28,29,30]. Using the fifth percentile as a cutoff point for low muscle strength and with GLIM criteria, the prevalence of malnutrition was 80.1%, almost the same as using SGA for malnutrition diagnosis.

Inflammation is an etiologic criterion in the GLIM classification, and it is widely accepted for both screening and nutritional assessment. Markers like serum albumin or CRP, used in the Glasgow prognostic score, are useful to detect inflammation. In our sample, 95% of patients presented a high Glasgow prognostic score, enabling its application as an etiologic criterion according to GLIM consensus. In addition, regardless of the used criterion for the detection and classification of malnutrition, albumin and prealbumin values were significantly lower and CRP higher than in normally nourished patients, consequently altering the Glasgow prognostic score. 

Even though all the tools we used in our study to classify malnutrition were associated with worse values regarding analysis (albumin, prealbumin, CRP, Glasgow prognostic score), weight (BMI), and length of hospital stay (approximately three more days), the only ones that were significantly associated with increased mortality in both the short and long term were SGA and GLIM criteria (using FFMI and HGS). In this sense, the best tools to predict six-moth mortality were SGA (odds ratio 2.73) and GLIM criteria using hand grip strength (odds ratio 2.72) or FFMI (odds ratio 1.87), even after adjusting for confounding variables like tumor stage and age. These findings support the use of SGA as a simple and inexpensive tool in the diagnosis of malnutrition in clinical routine. Bearing in mind the feasible standardization of hand grip in clinical practice (it is less bound to intra and inter professional variability than skinfold measurement) [31], we think that hand grip strength is a proper tool to apply muscle-related phenotypic criteria according to GLIM consensus. 

Our study has several strengths; it is a prospective study with a substantial number of subjects and with long-term monitoring. Furthermore, it applies simple techniques to measure and define reduced muscle mass, which can be useful when other methods are not available. 

All the same, there are potential limitations in our study. It was a single-center observational study; thus, results should be interpreted with caution and no causal links can be drawn. The inclusion of patients with a variety of tumor sites may have contributed to the heterogeneity of the population studied. The patients evaluated were hospitalized at the moment of the study and had an advanced-stage tumor, which is associated with an increased mortality. Nevertheless, this fact could reinforce the value of the assessed tools as they are capable of discriminating even among high-risk patients. 

In conclusion, the prevalence of malnutrition is very high among advanced-stage cancer inpatients. SGA and GLIM, especially with the use of hand grip strength to assess muscle-related criteria, are adequate tools to diagnose malnutrition and have a similar predictive value for six-month mortality in cancer patients. 

This study opens the path to perform further studies in different groups of patients to confirm the utility of this approach comparing these tools for muscle-mass assessing with other more complex and expensive approaches. 

## Figures and Tables

**Figure 1 nutrients-11-02043-f001:**
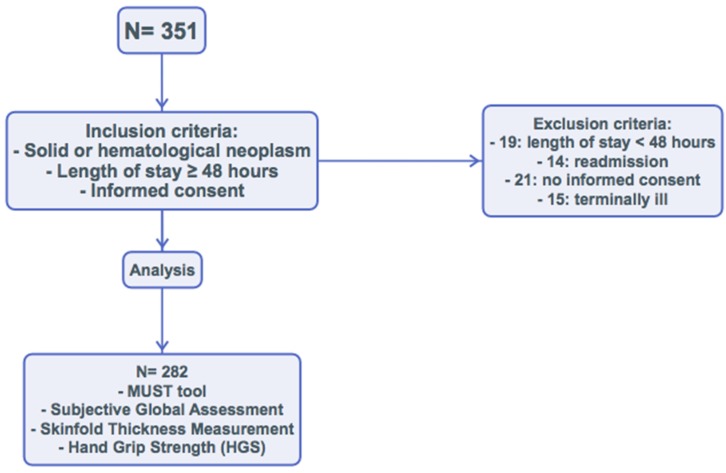
Flow chart.

**Figure 2 nutrients-11-02043-f002:**
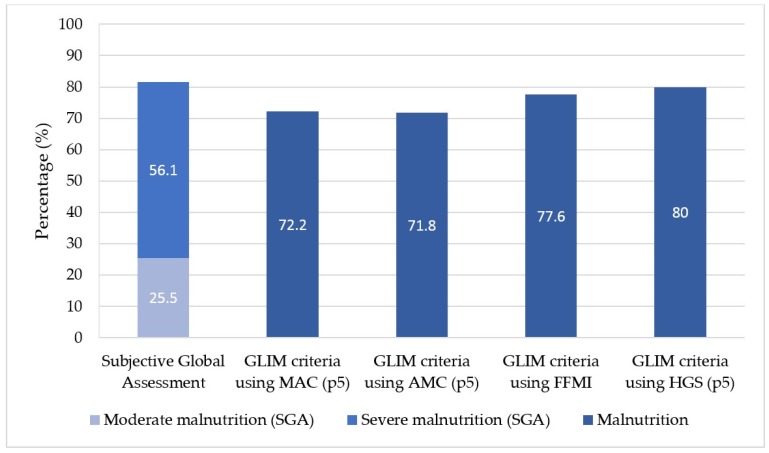
Malnutrition diagnosis according to the tool used. p5: fifth percentile.

**Table 1 nutrients-11-02043-t001:** General features.

		*n* = 282
**Age** (years)	mean ± SD	60.4 ± 12.6
**Sex**	*n* (%)	
Men		157 (55.7)
Women		125 (44.3)
**Type of admission**	*n* (%)	
Scheduled		51 (18.1)
Urgent		231 (81.9)
**Tumor stage**	*n* (%)	
I		7 (2.5)
II		13 (4.6)
III		50 (17.7)
IV		212 (75.2)
**Glasgow prognostic score**	*n* (%)	
No inflammation		13 (4.6)
Inflammation		269 (95.4)
**BMI** (kg/m^2^)	mean ± SD	
Men		24.7 ± 4.9
Women		24.5 ± 5.1
**Mid-arm circumference** (cm)	mean ± SD	
Men		26.6 ± 3.3
Women		26.3 ± 4.3
<p5	*n* (%)	39 (14.8)
**Arm muscle circumference** (cm)	mean ± SD	
Men		22.9 ± 2.7
Women		20.5 ± 2.8
<p5	*n* (%)	24 (9.1)
**Fat-free mass index** (kg/m^2^)	mean ± SD	
Men		17.49 ± 2.42
Women		15.93 ± 2.48
**Hand grip strength (kg)**	mean ± SD	
Men		26.21 ± 8.58
Women		16.51 ± 6.70
<p5	*n* (%)	107 (37.9)
**6-month death**	*n* (%)	135 (47.9)

BMI: body mass index; FFMI: fat-free mass index; SD: standard deviation; p5: fifth percentile.

**Table 2 nutrients-11-02043-t002:** Nutritional assessment.

		*n* = 282
**MUST tool**	*n* (%)	
0 points (low malnutrition risk)		48 (17.0)
1 point (medium malnutrition risk)		42 (14.9)
2 points or more (high malnutrition risk)		192 (68.1)
**Global subjective assessment**	*n* (%)	
Normally nourished		52 (18.4)
Moderate malnutrition		72 (25.5)
Severe malnutrition		158 (56.1)
**BMI**		
Low BMI	*n* (%)	58 (20.6)
Normal and high BMI		224 (79.4)
**FFMI**	*n* (%)	
Normal FFMI ^a^		67 (23.8)
Low FFMI ^a^		215 (76.2)
**Malnutrition by GLIM criteria**	*n* (%)	
**Using mid-arm circumference p5**		
Normally nourished		72 (27.8)
Malnutrition		187 (72.2)
**Using arm muscle circumference p5**		
Normally nourished		73 (28.2)
Malnutrition		186 (71.8)
**Using FFMI**		
Normally nourished		62 (22)
Malnutrition		215 (77.6)
**Using hand grip p5**		
Normally nourished		54 (20)
Malnutrition		216 (80)

^a^ ESPEN criteria: European Society for Clinical Nutrition and Metabolism criteria; MUST: malnutrition universal screening tool; BMI: body mass index; GLIM: Global Leadership Initiative on Malnutrition; FFMI: fat-free mass index; p5: fifth percentile.

**Table 3 nutrients-11-02043-t003:** Characteristics of patients admitted at medical oncology inpatient service, according to subjective global assessment and GLIM malnutrition criteria, according to different fat-free mass indexes.

	Subjetive Global Assessment			GLIM Criteria Using Mid-Arm Circumference (p5)		GLIM Criteria Using Arm Muscle Circumference (p5)		GLIM Criteria Using Grip Strength (p5)		GLIM Criteria Using FFMI by Anthropometry	
	Normally Nourished (*n* = 52)	Moderate Malnutrition (*n* = 71)	Severe Malnutrition (*n* = 157)	Normally Nourished (*n* = 72)	Malnourished (*n* = 187)	Normally Nourished (*n* = 77)	Malnourished (*n* = 197)	Normally Nourished (*n* = 54)	Malnourished (*n* = 216)	Normally Nourished (*n* = 62)	Malnourished (*n* = 215)
mean ± standard deviation											
**Age (years)**	57.9 ± 12.4	58.0 ± 14.6	62.3 ± 11.3 *	58.9 ± 12.8	61.5 ± 11.8	58.6 ± 12.7	61.7 ± 11.8	57.5 ± 11.8	61.5 ± 12.3 *	57.8 ± 13	61.5 ± 11.8 *
**Charlson index**	4.9 ± 2.1	4.9 ± 2.2	5.8 ± 1.9 *	5 ± 2.1	5.6 ± 2 *	4.9 ± 2	5.6 ± 2 *	4.7 ± 2.1	5.6 ± 2 *	4.8 ± 2.1	5.6 ± 2 *
**BMI (kg/m^2^)**	28.1 ± 3.7	25.4 ± 3.9	23.1 ± 5.1 *	27.2 ± 3.2	23.7 ± 5.3 *	27.1 ± 3.2	23.7 ± 5.3 *	27.0 ± 3.4	23.9 ± 5.1 *	27.8 ± 2.8	23.7 ± 5.1 *
**CRP (mg/dL)**	34.5 ± 58.5	69.3 ± 87.5	81.8 ± 90.3	51.4 ± 75.5	78.4 ± 91.1 *	50.1 ± 75.3	79.0 ± 91.1 *	49.1 ± 74.7	76.9 ± 89.4 *	54.4 ± 81.3	74.5 ± 87.7
**Albumin (g/dL)**	3.27 ± 0.58	2.81 ± 0.58	2.57 ± 0.59 *	3.06 ± 0.6	2.64 ± 0.61 *	3.07 ± 0.6	2.63 ± 0.61 *	3.14 ± 0.56	2.66 ± 0.63 *	3.05 ± 0.62	2.66 ± 0.62 *
**CRP/albumin ratio**	14.1 ± 26.7	30.4 ± 40.1	42.9 ± 77.5 *	22.1 ± 34.4	40.2 ± 74.7	21.5 ± 34.5	40.6 ± 74.6	20.0 ± 32.4	39.0 ± 71.1	23.7 ± 37	37.9 ± 70.6
**Prealbumin (mg/dL)**	24.1 ± 8.9	18.6 ± 8.4	14.9 ± 7.5 *	21.4 ± 9.8	15.9 ± 7.9 *	21.6 ± 9.7	15.8 ± 7.9 *	21.6 ± 8	16.1 ± 8.2 *	21.3 ± 9.9	16.2 ± 8 *
**Lymphocytes (× 10^9^)**	1.138 ± 0.654	1.030 ± 0.758	1.095 ± 0.695	1.099 ± 0.703	1.087 ± 0.709	1.117 ± 0.712	1.080 ± 0.705	1.091 ± 0.729	1.093 ± 0.707	1.107 ± 0.647	1.085 ± 0.72
**Length of stay (days)**	8.2 ± 9.8	8.1 ± 8.4	12.1 ± 8.1 *	8.6 ± 9.4	11.1 ± 7.9 *	8.5 ± 9.3	11.2 ± 7.9 *	7.7 ± 9.1	11.1 ± 8.1 *	8.4 ± 10	11.0 ± 7.7 *
*n* (%)											
**In-hospital death**	0 (0%)	3 (4.2%)	26 (16.6%) *	3 (4.2%)	22 (11.8%)	3 (3.9%)	22 (11.2%)	1 (1.9%)	25 (11.6%) *	2 (3.2%)	25 (11.6%) *
**6-month death**	16 (30.8%)	30 (42.2%)	88 (56.1%) *	27 (37.5%)	95 (50.8%)	29 (37.7%)	97 (49.2%)	16 (29.6%)	112 (51.9%) *	22 (35.5%)	111 (51.6%) *
**New admission (6-month)**	15 (28.8%)	35 (49.3%)	64 (40.8%)	26 (36.1%)	80 (42.8%)	28 (36.4%)	85 (46.1%)	17 (31.5%)	91 (42.1%)	23 (37.1%)	90 (41.9%)

BMI = body mass index; CRP= C-reactive protein; FFMI = fat-free mass index; * *p* < 0.05.

**Table 4 nutrients-11-02043-t004:** Association between malnutrition and mortality (six-month mortality risk).

	Crude				Adjusted			
	Odds Ratio	95% CI		*p*-Value	Odds Ratio	95% CI		*p*-Value
		Lower	Upper			Lower	Upper	
Subjective global assessment								
normally nourished vs. moderate malnutrition	1.65	0.78	3.5	0.328	1.48	0.67	3.26	0.24
Normally nourished vs. severe malnutrition	2.87	1.47	5.6	0.002	2.73	1.35	5.52	0.002
Normally nourished vs. malnutrition (2 groups)	2.41	1.27	4.6	0.007	2.23	1.14	4.38	0.009
Malnutrition according GLIM using mid-arm circumference (p5)	1.72	0.99	3.01	0.056	1.73	0.96	3.13	0.068
Malnutrition according GLIM using arm muscle circumference (p5)	1.61	0.937	2.75	0.085	1.73	0.97	3.1	0.064
Malnutrition according GLIM using FFMI	1.94	1.08	3.48	0.026	1.87	1.01	3.48	0.047
Malnutrition according GLIM using hand grip strength (p5)	2.56	1.35	4.86	0.004	2.72	1.37	5.4	0.004

Adjusted for age, sex and cancer stage. CI = confidence interval; SGA = subjective global assessment; FFMI = fat-free mass index.

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
