# Peer review of "GLIM Criteria Using Hand Grip Strength Adequately Predict Six-Month Mortality in Cancer Inpatients"

_nutrients, 2019, doi:10.3390/nu11092043_

Round 1

Reviewer 1 Report

Title: six-month, not sis

Recommendations (->): 

Brachial circumference -> mid-arm circumference;

Brachial muscle circumference -> arm muscle circumference;

38 line - odds ratio;

45 line - hangrip -> handgrip strenght or hand grip strenght;

51 line - not [1] [2] [3] but [1-4];

54 line - "are often" -> "are common";

58 line - [1] [6] [8] -> [1, 6, 8];

231 line - [33][34][35][36] ->[33-36];

65 line - Should be lowercase: "dual-energy x-ray..., bioelectrical...";

66 line - "magnetic resonance image"-> "magnetic resonance imaging";

69 line - "Skinfold Thickness Measurement" -> "skinfold thickness measurement";

69 line - "However, Skinfold Thickness Measurement (STM) is not mentioned as a useful method for body composition assessment" -> but skinfold thickness measurement is used for mid-arm muscle circumference calculation;

78-79 lines - Comma missing in "Our hypothesis is that GLIM criteria, using simple bedside-available tools for measuring muscle  functioncan adequately predict six-month mortality in cancer inpatients in a similar manner to other widely used tools";

86 line - "Hospital Regional Universitario de Málaga" - translate to English?

Figure 1 - MUST tool? Subjetive global assessment -> Subjective Global Assessment; exclusion criteria;

134 line - "Kolmogorof-Smirnof test" - >"Kolmogorov-Smirnof test";

151 line - "...detecting malnutrition risk in 234 patients (82.9%: 14.9% moderate risk y 68.1% high risk)" -> perhaps "detecting malnutrition risk in 82.9℅ (234) patients: 14.9℅ moderate risk and 68.1℅ high risk";

156 line - After six months, 135 patients (47.9%) had deceased" - >"After six months, 47.9% (135) patients had deceased";

128 line - Glasgow Prognostic Score, but in table 1: Glasgow prognostic score (you need to be consistent in your use of capital letters across the paper)

Table 1 - m ± DS -> m ± SD (standard deviation);

Table 1 - BMI: Body mass index. FFMI: Free fat mass index - > Fat free mass index; SD - standard deviation;

Table 2: MUST tool is malnutrition risk tool, has max. 7 points, how do you define moderate malnutrition? Also severe nutrition - > severe malnutrition. Maybe: moderate malnutrition risk?;

Table 2- Normo-nutrition? -> Normally nourished;

Table 3 - albumin value 3.27±0.58 and other is not right -> 32.7±5.8 g/L;

Table 3 - Lymphocytes (x910) value is not right -> Lymphocytes (x109); 1138±654 -> 1.138±0.654SI units: 5.4 x 109/L but not 5400 cells/mm3 or cells/mkL, for example;

Table 3 - In-hospital Death;

Table 3 - *p<0.05; **p<0.01; ***p<0.001 - it is excessive. Stay consistent - in line 139 you said: "For calculations, significance was set at p<0.05";

162 line - "Despite not having statistical significance, readmissions were higher among malnourished patients -> it is incorrect;

Table 1 - Type of admission Programmed -> planned, scheduled?;

Table 1 - 6-month Exitus? -> did you mean death?;

Table 2 - Using Had grip -> Hand grip;

Table 2 - what is the difference between "Malnutrition by FFMIa" and "Malnutrition by GLIM criteria: Using FFMI"?;

Table 4 title : "Table 4. Logistic regression analysis. Adjusted for age, sex and cancer stage" - needs to be formulated better;

Table 4 possible correction: Subjective global assessment Normally nourished vs Moderate malnutrition 0.24 (p value) and Subjective global assessment Normally nourished vs Severe malnutrition 0.002, and SGAa: normally nourished vs malnutrition (2 groups) 0,009;

aSGA=Subjective Global Assessment; bFFMI= Fat-Free Mass Index;

166 line - "significantly associated with malnutrition and severe malnutrition by SGA..." - > maybe "malnutrition (2 groups) and severe malnutrition..."

Almost all References must be corrected. A few examples:

34. [34] Kufeldt J, Viehrig M, Schweikert D, Fritsche A, Bamberg M, Adolph M. Treatment of malnutrition decreases complication rates and shortens the length of hospital stays in a radiation oncology department. Strahlentherapie Und Onkol Organ Der Dtsch Rontgengesellschaft . [et Al] 2018;194:1049–59. 372 doi:10.1007/s00066-018-1360-9.;

31. [31] Kilgour RD, Vigano A, Trutschnigg B, Lucar E, Borod M, Morais JA. Handgrip strength predicts survival and is associated with markers of clinical and functional outcomes in advanced cancer patients. Support Care Cancer Off J Multinatl AssocSupport Care Cancer 2013;21:3261–70. doi:10.1007/s00520-363 013-1894-4.

24. [24] Gioulbasanis I, Baracos VE, Giannousi Z, Xyrafas A, Martin L, Georgoulias V, et al. Baseline nutritional evaluation in metastatic lung cancer patients: Mini Nutritional Assessment versus weight loss history. Ann Oncol Off J Eur Soc Med Oncol 2011;22:835–41. doi:10.1093/annonc/mdq440.

35. [35] Planas M, Alvarez-Hernandez J, Leon-Sanz M, Celaya-Perez S, Araujo K, Garcia de Lorenzo A. Prevalence of hospital malnutrition in cancer patients: a sub-analysis of the PREDyCES(R) study. Support Care Cancer Off J Multinatl Assoc Support Care Cancer 2016;24:429–35. doi:10.1007/s00520-015-2813-7.

New references about GLIM criteria are:

Cederholm C, ...ESPEN Endorsed Recommendation GLIM criteria for the diagnosis of malnutrition e A consensus report from the global clinical nutrition community*. Clinical Nutrition 38 (2019) 1-9.

or 

Jensen GL, ...GLIM Criteria for the Diagnosis of Malnutrition:A Consensus Report From the Global ClinicalNutrition Community in JPEN, 2019.

Author Response

Dear Reviewer,

Thank you for giving us the opportunity to improve our article “GLIM Criteria Using Hand Grip Strength Adequately Predicts Six-Month Mortality in Cancer Inpatients”.

The various suggestions have been incorporated into the new version wherever applicable. Please find below our responses and the action taken to all the various suggestions and comments.

Please see the attachment to check the new version of the manuscript.

Once again, we very much appreciate all the work with the review.

Yours sincerely,

Dr. Gabriel Olveira

Dr. Victoria Contreras

Dr. Francisco José Sánchez Torralvo

R:

Comments and Suggestions for Authors

Title: six-month, not sis

Recommendations (->):

Brachial circumference -> mid-arm circumference;

Brachial muscle circumference -> arm muscle circumference;

38 line - odds ratio;

45 line - hangrip -> handgrip strenght or hand grip strenght;

51 line - not [1] [2] [3] but [1-4];

54 line - "are often" -> "are common";

58 line - [1] [6] [8] -> [1, 6, 8];

231 line - [33][34][35][36] ->[33-36];

65 line - Should be lowercase: "dual-energy x-ray..., bioelectrical...";

66 line - "magnetic resonance image"-> "magnetic resonance imaging";

69 line - "Skinfold Thickness Measurement" -> "skinfold thickness measurement";

G.O.: Thank you for your appreciation that improves our manuscript. All the suggestions above have been incorporated into the new version.

R:

69 line - "However, Skinfold Thickness Measurement (STM) is not mentioned as a useful method for body composition assessment" -> but skinfold thickness measurement is used for mid-arm muscle circumference calculation.

G.O.: Thank you for your appreciation that improves our manuscript. We have erased the wrong statement.

R:

78-79 lines - Comma missing in "Our hypothesis is that GLIM criteria, using simple bedside-available tools for measuring muscle  function, can adequately predict six-month mortality in cancer inpatients in a similar manner to other widely used tools";

G.O.: Thank you. The suggestion above has been incorporated into the new version.

R:

86 line - "Hospital Regional Universitario de Málaga" - translate to English?

G.O.: Thank you for your appreciation. We usually keep the name in Spanish.

R:

Figure 1 - MUST tool? Subjetive global assessment -> Subjective Global Assessment; exclusion criteria;

134 line - "Kolmogorof-Smirnof test" - >"Kolmogorov-Smirnof test";

151 line - "...detecting malnutrition risk in 234 patients (82.9%: 14.9% moderate risk y 68.1% high risk)" -> perhaps "detecting malnutrition risk in 82.9℅ (234) patients: 14.9℅ moderate risk and 68.1℅ high risk";

156 line - After six months, 135 patients (47.9%) had deceased" - >"After six months, 47.9% (135) patients had deceased";

128 line - Glasgow Prognostic Score, but in table 1: Glasgow prognostic score (you need to be consistent in your use of capital letters across the paper)

Table 1 - m ± DS -> m ± SD (standard deviation);

Table 1 - BMI: Body mass index. FFMI: Free fat mass index - > Fat free mass index; SD - standard deviation;

G.O.: Thank you for your appreciation that improves our manuscript. All the suggestions above have been incorporated into the new version.

R:

Table 2: MUST tool is malnutrition risk tool, has max. 7 points, how do you define moderate malnutrition? Also severe nutrition - > severe malnutrition. Maybe: moderate malnutrition risk?

G.O.: We very much appreciate your suggestion. We have ordered the options in table 2 as: "0 points - low risk -; 1 point -medium risk-; and 2 or more points -high risk-" as indicated in the interpretation of the MUST tool.

R:

Table 2- Normo-nutrition? -> Normally nourished;

Table 3 - albumin value 3.27±0.58 and other is not right -> 32.7±5.8 g/L;

Table 3 - Lymphocytes (x910) value is not right -> Lymphocytes (x109); 1138±654 -> 1.138±0.654. SI units: 5.4 x 109/L but not 5400 cells/mm3 or cells/mkL, for example;

Table 3 - In-hospital Death;

Table 3 - *p<0.05; **p<0.01; ***p<0.001 - it is excessive. Stay consistent - in line 139 you said: "For calculations, significance was set at p<0.05";

G.O.: Thank you for your appreciation that improves our manuscript. All the suggestions above have been incorporated into the new version.

R:

162 line - "Despite not having statistical significance, readmissions were higher among malnourished patients -> it is incorrect.

G.O.: Thank you for your appreciation that improves our manuscript. We have erased the wrong statement.

R:

Table 1 - Type of admission Programmed -> planned, scheduled?;

Table 1 - 6-month Exitus? -> did you mean death?;

Table 2 - Using Had grip -> Hand grip;

G.O.: Thank you for your appreciation. All the suggestions above have been incorporated into the new version.

R:

Table 2 - what is the difference between "Malnutrition by FFMIa" and "Malnutrition by GLIM criteria: Using FFMI"?

G.O.: Thank you for your appreciation that improves our manuscript. We have changed the classification in relation to the FFMI and we have ordered it as "normal FFMI" and "low FFMI" according to the ESPEN cut-off points.

R:

Table 4 title: "Table 4. Logistic regression analysis. Adjusted for age, sex and cancer stage" - needs to be formulated better.

G.O.: Thank you for your appreciation that improves our manuscript. We have change table 4 title.

R:

Table 4 possible correction: Subjective global assessment Normally nourished vs Moderate malnutrition 0.24 (p value) and Subjective global assessment Normally nourished vs Severe malnutrition 0.002, and SGAa: normally nourished vs malnutrition (2 groups) 0,009;

aSGA=Subjective Global Assessment; bFFMI= Fat-Free Mass Index;

G.O.: Thank you for your appreciation. We have used the suggested correction in table 4.

R:

166 line - "significantly associated with malnutrition and severe malnutrition by SGA..." - > maybe "malnutrition (2 groups) and severe malnutrition..."

G.O.: Thank you. The suggestion above has been incorporated into the new version.

R:

Almost all References must be corrected. A few examples:

[34] Kufeldt J, Viehrig M, Schweikert D, Fritsche A, Bamberg M, Adolph M. Treatment of malnutrition decreases complication rates and shortens the length of hospital stays in a radiation oncology department. Strahlentherapie Und Onkol Organ Der Dtsch Rontgengesellschaft . [et Al] 2018;194:1049–59. 372 doi:10.1007/s00066-018-1360-9.; [31] Kilgour RD, Vigano A, Trutschnigg B, Lucar E, Borod M, Morais JA. Handgrip strength predicts survival and is associated with markers of clinical and functional outcomes in advanced cancer patients. Support Care Cancer Off J Multinatl AssocSupport Care Cancer 2013;21:3261–70. doi:10.1007/s00520-363 013-1894-4. [24] Gioulbasanis I, Baracos VE, Giannousi Z, Xyrafas A, Martin L, Georgoulias V, et al. Baseline nutritional evaluation in metastatic lung cancer patients: Mini Nutritional Assessment versus weight loss history. Ann Oncol Off J Eur Soc Med Oncol 2011;22:835–41. doi:10.1093/annonc/mdq440. [35] Planas M, Alvarez-Hernandez J, Leon-Sanz M, Celaya-Perez S, Araujo K, Garcia de Lorenzo A. Prevalence of hospital malnutrition in cancer patients: a sub-analysis of the PREDyCES(R) study. Support Care Cancer Off J Multinatl Assoc Support Care Cancer 2016;24:429–35. doi:10.1007/s00520-015-2813-7.

New references about GLIM criteria are:

Cederholm C, ...ESPEN Endorsed Recommendation GLIM criteria for the diagnosis of malnutrition e A consensus report from the global clinical nutrition community*. Clinical Nutrition 38 (2019) 1-9.

or

Jensen GL, ...GLIM Criteria for the Diagnosis of Malnutrition:A Consensus Report From the Global ClinicalNutrition Community in JPEN, 2019.

G.O.: We very much appreciate your suggestion. We have reviewed the bibliographic references in depth and we have solved the errors detected.

Reviewer 2 Report

Your manuscript is very important and interesting. I think it is an excellent idea to test muscle strength along with malnutrition tools.

Author Response

Dear Reviewer,

Thank you for giving us your opinion about our article “GLIM Criteria Using Hand Grip Strength Adequately Predicts Six-Month Mortality in Cancer Inpatients”.

We very much appreciate it. 

We are very pleased with the interest that you have expressed.

Yours sincerely,

Dr. Gabriel Olveira

Dr. Victoria Contreras

Dr. Francisco José Sánchez Torralvo

Reviewer 3 Report

Summary

This is an interesting as well as relevant topic discussing the usefulness of the GLIM criteria in mortality-prediction of cancer patients and comparing the GLIM criteria to other tools, such as the Subjective Global Assessment and Handgrip Strength.

The quality of the English must be improved to become acceptable for publication.

The tables are informative, but the formatting needs to be revised. There are no figures evaluating the findings of this study.

This manuscript needs to be structured. Many parts are repetitive.

Overall, while the topic is very interesting and relevant and would add to the knowledge in this field, the important aspects were not covered in detail and the manuscript does not appear to be prepared with the necessary thoroughness.

Broad comments 

An extensive English revision is needed to become acceptable for publication. The writing style is very plain, spelling and grammar errors are frequent.

Please stick with one description of nutritional state and one formatting style of tables in your entire manuscript.

Please introduce all abbreviations once and only if used in the manuscript more than twice.

Please introduce all applied and discussed screening and assessment tools and equipments

I strongly advocate that the concepts of malnutrition, muscle mass, muscle strength and physical function should not be mixed and for each of which different diagnostic criteria should be applied

Specific comments

The introduction needs some structural changes to lead from malnutrition in cancer patients to the cinical symptoms and to assessment and evaluation possibilities to the recently published GLIM criteria compared to existing tools. Methods: the MUST was not introduced Results: half of the results part is only the description of the baseline characteristics, special attention should be paid to the different criteria of the malnutrition tools used. These tools have to be compared more closely to be useful. What impact did the results of HGS, MUST, SGA and GLIM have on patient outcome? This should not only be presented in tables without description Discussion: lines 180-224 are a mixture of background and results, there is a lack of structure and data interpretation Pg 1, line 24: please rephrase Pg 1, line 26: please rephrase Pg 2, line 79: the GLIM criteria do not assess muscle function and do not include muscle strength Pg 2 and 3, lines 85-92: exclusion criteria of figure and text are not identical Pg 8, line 213: p5 was not introduced Pg 8, line 225: the Glasgow index was not introduced or reported on Pg 9, lines 249: I disagree on HGS being easy to use. The authors own data with large standard deviations add to that.

Author Response

Dear Reviewer,

Thank you for giving us the opportunity to improve our article “GLIM Criteria Using Hand Grip Strength Adequately Predicts Six-Month Mortality in Cancer Inpatients”.

The various suggestions have been incorporated into the new version wherever applicable. Please find below our responses and the action taken to all the various suggestions and comments.

Please see the attachment to check the new version of the manuscript.

Once again, we very much appreciate all the work with the review.

Yours sincerely,

Dr. Gabriel Olveira

Dr. Victoria Contreras

Dr. Francisco José Sánchez Torralvo

R:

Summary

This is an interesting as well as relevant topic discussing the usefulness of the GLIM criteria in mortality-prediction of cancer patients and comparing the GLIM criteria to other tools, such as the Subjective Global Assessment and Handgrip Strength

The quality of the English must be improved to become acceptable for publication

G.O.: We very much appreciate your suggestion in order to improve our manuscript. We have carried out an in-depth review of the text, trying to correct the grammatical and spelling errors.

R:

The tables are informative, but the formatting needs to be revised. There are no figures evaluating the findings of this study.

G.O.: We very much appreciate your suggestion. We have carried out an in-depth review of the tables.  

In the other hand, we have not added any new figures because we have not found a design that brings something new to the information shown in the tables. However, we will be pleased to evaluate new suggestions.

R:

This manuscript needs to be structured. Many parts are repetitive.

G.O.: Thank you for your appreciation that improves our manuscript. We have made several structural changes in the introduction, material and methods, results and discussion section in the new version of the manuscript.

R:

Overall, while the topic is very interesting and relevant and would add to the knowledge in this field, the important aspects were not covered in detail and the manuscript does not appear to be prepared with the necessary thoroughness.

G.O.: We very much appreciate your opinion. We have made several structural changes in the whole manuscript. In any case, we strongly encourage you to point out the changes that you still consider necessary to be made

R:

Broad comments

An extensive English revision is needed to become acceptable for publication. The writing style is very plain, spelling and grammar errors are frequent.

G.O.: We very much appreciate your suggestion in order to improve our manuscript. We have carried out an in-depth review of the text, trying to correct the grammatical and spelling errors.

R:

Please stick with one description of nutritional state and one formatting style of tables in your entire manuscript.

G.O.: We very much appreciate your suggestion. We have tried to homogenize the contents of the tables, with special emphasis on the description of the nutritional status.

R:

Please introduce all abbreviations once and only if used in the manuscript more than twice.

G.O.: Thank you for your appreciation. We have revised all the abbreviations.

R:

Please introduce all applied and discussed screening and assessment tools and equipments

G.O.: Thank you for your appreciation that improves our manuscript. We have expanded the material and methods section including all the screening and assessment tools and equipments.

R:

I strongly advocate that the concepts of malnutrition, muscle mass, muscle strength and physical function should not be mixed and for each of which different diagnostic criteria should be applied.

G.O.: We very much appreciate your suggestion. We have tried to improve the use of these terms with the corrections we have made in the new version.

R:

Specific comments

The introduction needs some structural changes to lead from malnutrition in cancer patients to the clinical symptoms and to assessment and evaluation possibilities to the recently published GLIM criteria compared to existing tools.

G.O.: Thank you for your appreciation that improves our manuscript. We have tried to improve the writing of the text in the new version.

R:

Methods: the MUST was not introduced.

G.O.: Thank you for your appreciation. We have expanded the material and methods section including all the screening and assessment tools and equipments.

R:

Results: half of the results part is only the description of the baseline characteristics, special attention should be paid to the different criteria of the malnutrition tools used. These tools have to be compared more closely to be useful. What impact did the results of HGS, MUST, SGA and GLIM have on patient outcome? This should not only be presented in tables without description.

G.O.: We very much appreciate your suggestion. We have changed the results section in order to present the results of the malnutrition tools used.

R:

Discussion: lines 180-224 are a mixture of background and results, there is a lack of structure and data interpretation

G.O.: Thank you for your appreciation that improves our manuscript.  We have changed those lines to improve data interpretation.

R:

Pg 1, line 24: please rephrase

Pg 1, line 26: please rephrase

G.O.: Thank you. We have rephrased those lines.

R:

Pg 2, line 79: the GLIM criteria do not assess muscle function and do not include muscle strength

G.O.: Thank you for your appreciation. We have rephrased that line.

R:

Pg 2 and 3, lines 85-92: exclusion criteria of figure and text are not identical

G.O.: Thank you for your appreciation. We have changed the phrase to correctly list the exclusion criteria.

R:

Pg 8, line 213: p5 was not introduced

Pg 8, line 225: the Glasgow index was not introduced or reported

G.O.: Thank you for your appreciation that improves our manuscript. We have introduced p5 and Glasgow Prognostic Score in the material and methods section.

R:

Pg 9, lines 249: I disagree on HGS being easy to use. The authors own data with large standard deviations add to that.

G.O.: We very much appreciate your suggestion in order to improve our manuscript. There was an incorrect expression. We have changed it.

Round 2

Reviewer 3 Report

Summary

This manuscript has been improved. Minor revisions are necessary to become acceptable in my opinion.

Broad comments 

The level of English has been improved to a good quality, there are few grammatical errors and typos left.

Minor formatting issues remain, including the used materials and unwanted line  and page breaks, the consistent use of abbreviations

I suggest to present at least one figure, displaying the detection rate of malnutrition with the different tools used (MUST, SGA, GLIM, GLIM+HGS, perhaps your laboratory markers, etc.)

Specific comments

Line 3: “predict” since “criteria” and “HGS” are plural Line 28 “their association” Line 29: redundant sentence Line 54: Please add that inflammation is associated with cancer Line 97: eliminate the second “performed” Line 99: consider “first diagnosis” if applicable Line 102: revise language Line 115: Please briefly describe what the Glasgow Prognostic Score is Line 155: adapt to your new abbreviations Line 191: from the background I think you mean GLIM criteria+ HGS -please specify Table 1: please introduce the abbreviations “m” in the legend, please specify which “stage” Lines 214-217, lines 230-23, lines 243-249: I would still consider omitting these redundant sections Lines 219 & 220: redundant Line 223: please rephrase, the SGA itself is not associated with quality of life etc. Line 255: please add “in other studies” Lines 262-269: in my opinion, this section should be moved to the introduction Line 270: please rephrase Line 292: this an important finding, inferring that the GLIM criteria only predict as well as the SGA, if the HGS and FFM are determined as well- which is difficult to apply in clinical routine, please discuss the implications for clinical routine Line 296: consider mentioning Bohannon et al (doi: 10.14283/jfa.2017.8.) ,and something is wrong with this reference Line 297: consider rephrasing “to apply muscle related GLIM criteria” Line 300: this part is unclear to me

Author Response

Dear Reviewer,

Thank you again for giving us the opportunity to improve our article “GLIM Criteria Using Hand Grip Strength Adequately Predict Six-Month Mortality in Cancer Inpatients”.

Most of the suggestions have been incorporated into the new version wherever applicable. Please find below our responses and the action taken to all the suggestions and comments.

Please see the attachment to check the new version of the manuscript.

Once again, we very much appreciate all the work with the review.

Yours sincerely,

Dr. Gabriel Olveira

Dr. Victoria Contreras

Dr. Francisco José Sánchez Torralvo

R: Broad comments

The level of English has been improved to a good quality, there are few grammatical errors and typos left.

G.O.: Thank you. We have carried out a new review of the text, trying to correct the grammatical errors.

R: Minor formatting issues remain, including the used materials and unwanted line and page breaks, the consistent use of abbreviations

G.O.: We have carried out an in-depth review of that formatting issues. Thank you for the appreciation.

R:

I suggest to present at least one figure, displaying the detection rate of malnutrition with the different tools used (MUST, SGA, GLIM, GLIM+HGS, perhaps your laboratory markers, etc.)

G.O.: We have made a new figure including the detection rate of malnutrition with the different tools used. Thank you for specifying the suggestion in this review.

R:

Specific comments

Line 3: “predict” since “criteria” and “HGS” are plural

Line 28 “their association”

Line 29: redundant sentence

Line 54: Please add that inflammation is associated with cancer

Line 97: eliminate the second “performed”

G.O.: Thank you for your particular appreciations that improve our manuscript. All the suggestions above have been incorporated into the new version. Lines 29 and 54 have been changed.

R:

Line 99: consider “first diagnosis” if applicable

G.O.: Thank you for the appreciation. In this case, we do not consider it applicable.

R:

Line 102: revise language

Line 115: Please briefly describe what the Glasgow Prognostic Score is

Line 155: adapt to your new abbreviations

Line 191: from the background I think you mean GLIM criteria+ HGS -please specify

G.O.: Thank you for your appreciations that improve our manuscript. All the suggestions above have been incorporated into the new version.

R:

Table 1: please introduce the abbreviations “m” in the legend, please specify which “stage”

G.O.: Thank you for your appreciation. We have changed and used the term "mean" in tables 1 and 2, as we did before in table 3. We now specify "tumor stage".

R:

Lines 214-217

lines 230-23

lines 243-249: I would still consider omitting these redundant sections (simplified)

G.O.: We very much appreciate your suggestions. We have omitted the first section and simplified and tried to improve the third. We consider that  section in lines 230-233 is used as an introduction to rest of the paragraph.

R:

Lines 219 & 220: redundant Line

223: please rephrase, the SGA itself is not associated with quality of life etc.

Line 255: please add “in other studies”

Lines 262-269: in my opinion, this section should be moved to the introduction

Line 270: please rephrase

G.O.: We very much appreciate your suggestions. We have changed redundant lines and rephrased the indicated lines. We moved to the introduction and improved most of the content on lines 262-269

R:

Line 292: this an important finding, inferring that the GLIM criteria only predict as well as the SGA, if the HGS and FFM are determined as well- which is difficult to apply in clinical routine, please discuss the implications for clinical routine

G.O.: Thank you for your appreciations that improve our manuscript. We have completed this paragraph.

R:

Line 296: consider mentioning Bohannon et al (doi: 10.14283/jfa.2017.8.) ,and something is wrong with this reference

G.O.: We very much appreciate your suggestion. We have used the recommended reference. Indeed, the previous reference was wrong.

R:

Line 297: consider rephrasing “to apply muscle related GLIM criteria”

G.O.: Thank you. We have changed the line.

R:

Line 300: this part is unclear to me

G.O.: Thank you for your appreciation. In fact, The phrase was not well structured and was confusing. we have changed it
